# Investigations on Fatigue Life of Tube Connections Based on International Codes of Pressure Vessel

**DOI:** 10.3390/ma16010231

**Published:** 2022-12-27

**Authors:** Wenxian Su, Qinqin Cao, Gaoyu Cui, Zhiwei Chen

**Affiliations:** 1School of Energy and Power Engineering, University of Shanghai for Science and Technology, Shanghai 200093, China; 2China Special Equipment Inspection and Research Institute (CSEI), Beijing 100029, China

**Keywords:** tube connections, fatigue life, FEM, ASME VIII-2, EN 13445

## Abstract

The fatigue assessment of tube connections under cyclic pressure is discussed using four kinds of methods from ASME VIII-2 and EN 13445-3. FEA results are compared to the fatigue test, and some conclusions are obtained. Method 1 is the most widely used traditional method and can be used in both welded structures and unwelded structures. This method has simple operation, safety and reliability. Method 2 adopts the effective strain range to assess the fatigue for both the welded and the unwelded structure. This method is with high accuracy, good stability, safety and reliability, but the elastic–plastic analysis is very complicated. Method 3 adopts the equivalent structure stress to assess the fatigue of the welded, it is developed from fracture mechanics, and the procedure is also very complicated. Method 4 is a detailed assessment procedure for the welded and unwelded, and it is the most accurate, stable and reliable among the four methods.

## 1. Introduction

Fatigue failure is one of the main failure modes of pressure vessels due to cyclic loading, and the common failure of the pressure vessel is mainly manifested by strength, stiffness, stability and corrosion failure. How to predict and solve the failure of pressure vessels has been a wide concern. Xiao, H. et al. [1] studied the post-buckling damage of thin plates under the plane compression load and predicted the fatigue life of plates by combining the coupling effect of buckling and fatigue damage. It provides a new method and practical means for considering the post-buckling damage and fatigue life analysis of engineering structures. Perce, S.O. et al. [2] found that the bellows with braided layers lead to sudden failure by applying circulating pressure. Ma, K. et al. [3] analyzed the fatigue growth behavior of coplanar cracks and fatigue life of 4130X steel hydrogen storage vessel by experimental and numerical simulation, indicating that the results are consistent with that in BS 7910 and ASME BPVC. XI.

Fatigue is very important in the design of pressure vessels. Mayer, H. et al. [4] discussed the main fatigue design parameters and evaluated the validity of these parameters over the range of fatigue conditions to be predicted. Niu, X.P. et al. [5] proposed three conversion models of fatigue for reactor pressure vessels (RPV) based on the stress-strength, load-life and strength damage interference theories. Giglio, M. [6] compared two nozzle types of pressure vessels under cyclic pressure according to ASME and VSR 1995 by numerical and experiments. Rudolph, J. et al. [7] presented a method called the local strain approach to assessing the fatigue life for cylinder-to-cylinder intersections and butt weld joints, finding that the derivation of fatigue curves for special weld details is shown under the aspect of practical application. Okrajni, J. et al. [8] studied thermo-mechanical fatigue of power plant components and provided design methods for highly reliable pressure vessels. Margolin, B.Z. et al. [9] formulated the local strain–stress criterion about fatigue failure for nuclear pressure vessel steels under low cyclic loading and predicted the fatigue life for triaxial stress state and non-stationary loading with regard to the maximum stress effect. Mashiri, F.R. et al. [10] studied the fatigue of tube-to-plate T joints under cyclic in-plane bending using the classification method. Fatigue tests were carried out on forty-eight specimens made of square hollow tubes welded to plates, considering the effects of in-line galvanizing, steel type, stress ratio and tube wall thickness.

Tube connections are one of the most common welded structures in pressure vessels. Welding defects may vary as local surface notches at the weld, weld bead roughness, weld ripples, local undercut, local shrinkage grooves, local root concavity, welding stop/start craters, etc. These inherent fatigue crack sources can shorten the fatigue fracture process by skipping the fatigue crack initiation stage. Furthermore, there are serious stress concentrations and high welding residual stress at the welding points, which will increase the risk of fatigue failure. Therefore, opening tubing connection structures with welding joints were chosen to study fatigue failure [11].

In recent years, the advanced design concepts and research results are given in codes of pressure vessels for fatigue design, such as ASME VIII-2, Rules for Construction of Pressure Vessels Division 2, Alternative Rules [12] and EN 13445 Unfired Pressure Vessels [13]. The design pressure in ASME VIII-2 ranges from 103 KPa to 68.95 MPa, while EN 13445 is greater than 0.05 MPa and is not limited to the maximum value. Two methods of a simplified and detailed assessment of fatigue life are provided in EN 13,445, while there are three methods of fatigue design in ASME VIII-2. However, because of the differences between fatigue assessment methods in the two codes, investigations and comparisons of the fatigue design methods in these two codes are of great significance [14].

This paper aims to compare some kinds of fatigue assessment methods according to ASME VIII-2 and EN 13445 by FE modeling combined with the fatigue test data of pipe connection structures in WRC Bulletin 335 [15].

## 2. Current Approaches to Fatigue Life Assessment in Codes

The fatigue design is in Section 5.5 of ASME VIII-2, named “Protection against failure from cyclic loading”. It includes three detailed fatigue design methods: elastic stress analysis (equivalent stress), elastic–plastic stress analysis (equivalent strain) and elastic stress analysis (structural stress). These three methods will be referred to as Method 1, Method 2 and Method 3, respectively, hereinafter. First, screening criteria were used to determine if fatigue analysis is required as part of a design. If the component does not satisfy the screening criteria, a fatigue evaluation is performed using the techniques. There are two fatigue design methods in EN 13445-3, which are the simplified assessment in Chapter 17 and the detailed assessment in Chapter 18. As for the detailed assessment of the fatigue life, it provides detailed assessment methods for the weldments, unweldments and bolts by the detailed classification of the stress. The detailed assessment, which is hereinafter referred to as Method 4, is a kind of universal method with a wider range of applications compared to the simplified assessment.

The ASME method needs to be designed to understand the peak alternating stress intensity, the peak local value of the amplitude of fluctuation of the Tresca stress intensity. Based on the strain cycling experimental data of the non-welded plate, it was compared with the material performance data. Of particular note is the requirement to know the fluctuations of all three principal stresses, and these fluctuations require knowledge of the local location of the peak stress intensity. Finite element analysis can be used to study the geometry of the known machined shape, but the localized peaks cannot be determined for the weld with the same geometric shape variation and the toe of the unmachined weld, and there is damage caused by stress concentration for the toe of the weld. Therefore, the material properties method used in ASME applies to geometry set far away from the weld but not to weld details.

### 2.1. Elastic Stress Analysis and Equivalent Stresses-Method 1

An effective total equivalent stress amplitude is used to evaluate the fatigue damage for results obtained from a linear elastic stress analysis. The controlling stress for the fatigue evaluation is the effective total equivalent stress amplitude, defined as one-half of the effective total equivalent stress range (P_L_ + P_b_ + Q + F) calculated for each cycle in the loading histogram.

The primary plus secondary plus peak equivalent stress (see Figure 1) is the equivalent stress, derived from the highest value across the thickness of a section, of the combination of all primary, secondary and peak stresses produced by specified operating pressures and other mechanical loads and by general and local thermal effects and including the effects of gross and local structural discontinuities. Examples of load case combinations for this stress category for typical pressure vessel components are shown in Table 1. The parameters are defined in Table 1, and Table 2 defines the parameters of Table 1.
(1)$$S_{a}=P_{l}+P_{b}+Q+F$$

### 2.2. Elastic–Plastic Stress Analysis and Equivalent Strains-Method 2

The effective strain range is used to evaluate the fatigue damage for results obtained from an elastic–plastic stress analysis. The effective strain range is calculated for each cycle in the loading histogram using either cycle-by-cycle analysis or the twice yield method. For the cycle-by-cycle analysis, a cyclic plasticity algorithm with kinematic hardening shall be used.

### 2.3. Fatigue Assessment of Welds—Elastic Analysis and Structural Stress—Method 3

An equivalent structural stress range parameter is used to evaluate the fatigue damage for results obtained from a linear elastic stress analysis. The controlling stress for the fatigue evaluation is the structural stress that is a function of the membrane and bending stresses normal to the hypothetical crack plane. This method is recommended for the evaluation of welded joints that have not been machined to a smooth profile.

Fatigue cracks at pressure vessel welds are typically located at the toe of a weld. For as-welded and weld joints subject to post-weld heat treatment, the expected orientation of a fatigue crack is along the weld toe in the through-thickness direction, and the structural stress normal to the expected crack is the stress measure used to correlate fatigue life data. For fillet welded components, fatigue cracking may occur at the toe of the fillet weld or the weld throat, and both locations shall be considered in the assessment.

### 2.4. Detailed Assessment of Fatigue Life-Method 4

Method 4 is presented in EN13445. It is a very modern one, taking into account the fact that welded regions show a different cyclic fatigue behavior than unwelded regions. In unwelded regions, a large proportion of the cyclic life is required for crack initiation and only a short proportion for crack propagation until the breakthrough of the crack or until rupture. In welded regions, in contrast, the existence of microcracks (crack-like weld defects of microscopic scale) has to be taken into consideration. For the weld directly loaded or partially permeable, Equation (2) is used to calculate the stress range at the weld throat. For other forms of welded parts, the structural stress near the hot spot for fatigue evaluation is obtained by extrapolation.
(2)$$\Delta \sigma ={\left(\sigma_{w}^{2}+\tau_{w}^{2}\right)}^{1/2}$$
where *σ*_w_ is the normal stress range at the weld throat, and *τ*_w_ is the tangential stress range at the weld throat.

## 3. Differences and Similarities

The differences and similarities of the design principles, the assessment parameters, and the fatigue curves between the above four fatigue assessment methods are as follows:

### 3.1. Design Principles and Assessment Parameters

The stress involved in all the fatigue design methods mainly refers to notch stress, structural stress, nominal stress and hot-spot stress. The so-called nominal stress is the stress away from the discontinuous area. The structural stress distributes linearly along the thickness direction, including the discontinuous effect of the whole structure, except the discontinuous effect of the notch. The notch stress refers to the overall stress of the notch root, including the nonlinear parts of the stress distribution. The hot-spot stress refers to the stress in the area with crack initiation potential, which is generally calculated by certain extrapolation methods. The types and locations of all the stresses are listed in Figure 2. The analysis of the notch stress is applied to Method 1 and Method 2, while the analysis of the structural stress is applied to Method 3 for the fatigue assessment of the weldments. There are differences between ASME and EN 13445 in the -definition of structural stress. In ASME, the structural stress is expressed as the function of the membrane stress and the bending stress perpendicular to the hypothetical potential crack, while in EN, it is defined as the sum of the primary stress and the secondary stress. For Method 4, the notch stress is applied to the fatigue assessment of the non-weldments and the bolts, while the hot-spot stress is applied to the fatigue assessment of the weldments. Method 1 and Method 2 in ASME VIII adopt half of the stress amplitude in the fatigue assessment, while Method 3 and Method 4 of EN 13445 adopt the full stress amplitude, namely, the stress range used in the fatigue assessment. For engineering components, it is necessary to design gaps and similar discontinuous structures, and due to the influence of discontinuous effects and stress concentration in structural discontinuities, such as near the root of the notch, there is high peak stress, and a local high-stress zone is formed. The greater the tensile limit of the material, the greater the weakening of the fatigue strength of the notch. Therefore, stress concentration should be avoided as much as possible in the design, and notch structure design should avoid the use of notch-sensitive materials as much as possible.

### 3.2. Fatigue Design Curve

For the fatigue design curve, the X coordinate is the allowable cycle number, and the Y coordinate is the stress amplitude or the stress range, which are, respectively, represented by the Curve *S-N* and Δσ-*N* Curve [16]. There are corresponding fatigue design curves for all four methods.

The fatigue curve in ASME VIII-2 has two forms. One is based on the fatigue curve of the smooth bar specimen, which is used in both Method 1 and Method 2, and the other is based on the fatigue curve obtained from the sample test of welded parts, which is adopted in Method 3. The fatigue curve of the smooth bar can be used for components welded or unwelded, while the fatigue curve of welded connections is only used for welded connections. The applicability is that the fatigue curve of a smooth bar is limited by the maximum number of cycles on the curve, but the fatigue curve of a welded connection is not because there is no endurance limit. In addition, the fatigue curve of the smooth bar has adjusted the maximum possible impact on the average stress and strain, while the fatigue curve of the welded does not adjust the impact on the fatigue life caused by the average tensile stress and component size. However, the thickness and average stress are considered when calculating the structural stress range in Method 3.

In EN 13445, the fatigue curves are defined, respectively, according to the welding components, the unwelded components and the bolts. Different from Method 1 and Method 2 in ASME VIII-2, the Y coordinate of the curve is the stress range instead of the stress amplitude. Furthermore, the safety factor values in the two systems are different, where the safety factors are 1.5 for the stress range (Y coordinate) and 10 for the cycle number (X coordinate) in EN 13445, while in ASME VIII-2, the safety factors are 2 for the stress amplitude and 20 for the cycle number.

### 3.3. Welding Material

The different fatigue design methods are applied to the different materials. Methods 1 and 2 are applicable to the welded and unwelded components. When Method 1 is applied to the fatigue assessment, if the influence of the partial notch or the welding joint is not calculated in the model, the fatigue strength weakened coefficient K_f_ will be introduced to magnify the stress amplitude based on the treatment conditions and the nondestructive testing situation of the specific partial notch or welding joint. However, this requirement is not referred to in Method 2. According to Kalnins et al. [17], the total stress amplitude assessment, including the peak stress, should be applied to the fatigue analysis. Therefore, the influence of the partial notch or the welding joint in the model should be calculated in Method 2.

Method 3 is a fatigue assessment method for the welded structure by referring to the EN. According to the standard, this method needs the permission of the asset owner or the user. It also suggests that Method 3 should be used for the analysis of the welded joint with non-smooth contours and Method 1 or Method 2 for the assessment of the welding joint with smooth contours. According to Method 3, sensitivity analysis should be used for the analysis of the dimension-changing welding throat.

In Method 4, there are different fatigue assessment methods used for the materials of the welded components, the unwelded components and the bolts. The standard defines three approaches to calculating the stress range of the welded components. As for the simple accessories, outside the area of the gross structural discontinuity and the straight and smooth butt weld, the nominal stress should be applied to the fatigue analysis. As for the weld directly under loads or partial penetration weld, the stress range of the weld throat should be calculated. As for the other weldments, the structural stress of the area near the hot spots should be obtained by extrapolation and applied to the fatigue assessment. In Method 4, the welded components are classified in detail. It suggests that the corresponding fatigue design curves be chosen according to the types of welds (including the full-weld, partial weld and filler weld), the forms of the components, the assessing position and the detection conditions of the weld.

### 3.4. Correction Coefficient

(a)Plastic correction. In Method 1, the fatigue loss coefficient *Ke* is introduced to consider the influence of the plastic. Method 2 is about the elastic–plastic analysis. The stable cycle stress–strain curve is applied to analyze the material property, and the plastic correction coefficient is no longer considered because the influence of the material in the plastic stage has been calculated. In Method 3, the Formulas are determined by the Neubers principle and the stress–strain curve model of the material hysteretic property, which is applied to calculate the corresponding local nonlinear structural stress and strain range. On this basis, the calculated structural stress range should be corrected for low cycle fatigue. According to Method 4, for both the weld and the unwelded, if the calculated nominal elastic structural stress range exceeds twice the yield stress of the material, the plastic correction coefficient *Ke* (referring to the mechanical load) or *Kν* (referring to the thermal load) should be considered. For the results obtained by the elastic–plastic analysis in Method 4, if the total strain range (including the elastic and plastic strain) caused by all the loads is known, then the plastic correction is unnecessary.(b)Thickness correction. In Methods 1 and 2, the fatigue curves of the smooth specimens are applied, so the thickness correction has been included in the curves. In Method 3, the corresponding parameter is applied to the formula to analyze the influence of the thickness. Method 4 defines the calculation method of the thickness correction coefficient for the weldments, unweldments and bolts. The formula of the thickness correction coefficient of the non-weldments involves the unknown allowable load cycle, so the calculation needs a hypothesis for the iterative computations, which greatly increases the complexity of the computation [18].(c)Temperature correction. There are usually differences between the design temperature of the vessel and the test temperature of the fatigue design curve, which will influence the assessment of the actual allowable life. In the ASME VIII-2, based on the ratio between the elasticity modulus of the specimen material at the test temperature (the environment temperature) of the design fatigue curve and the elasticity modulus of the specimen material at the design temperature, the allowable cycle index is adjusted. Method 4 determines the calculation method of the coefficient for temperature correction to the weldments, non-weldments and bolts. In the design, we should consider that the strength of the metal at high temperatures is reduced, and creep failure may be caused when the temperature is too high; low temperature can improve the fatigue strength of metal materials to a certain extent, but at the same time, it will also reduce the toughness of the material and increase the brittleness, so too-high temperature and too-low temperature are harmful to fatigue.(d)Mean stress correction. In Method 1 and Method 2, the fatigue curve of the smooth specimen is adopted, and the mean stress correction is included in the curves. In Method 3, the relevant parameter is applied to evaluate the influence of the thickness in the formula. Similar to Method 1 and Method 2, the influence of the mean stress is involved in the fatigue design curve of the weld in Method 4; however, for the unweld, the corrections are made separately according to the absolute value of the maximum stress and the range of stress. It uses the full mean stress correction under the purely elastic state and the reducing mean stress correction under the shakedown elastic–plastic state. Under the cyclic plastic state, the plastic correction is considered instead of the mean stress correction. The mean stress correction coefficient is related to the yield strength, the tensile strength, the maximum stress, the stress range, the mean stress, the allowable cycle number, etc., where the initial value for the iteration needs to be assumed.(e)Surface treatment correction. For the three methods in ASME VIII-2, the correction factor of 20 for the number of cycles accounts for factors that actually affect the fatigue life but have not been considered in tests, including surface correction factors. In Method 4, the smooth standard specimen is applied to the assessment of the unweld, but it does not resemble the structure in the real world. Thus, the correction coefficient of the surface roughness is introduced to the assessment. Similar to the calculation of the thickness correction coefficient, because of the unknown allowable cycle number, iterative computation is required in the assessment, which makes the computation more complex [19].

### 3.5. Histogram Development and Cycle Counting

ASME VIII-2 provides two methods for determining the law of loading, namely the rain flow method and the max–min cycle counting method: (1) The rain flow cycle counting method is equivalent to a closed stress–strain hysteresis curve, and one hysteresis represents one cycle. When the change in load, stress or strain with time can be represented by a single parameter, the rain flow counting method is recommended. (2) The max–min cycle counting method is recommended to determine the point in time that represents a single cycle under non-proportional load. Cycle counting is performed by first constructing the largest possible cycle, using the highest and lowest valley values, then the second largest cycle, and so on until all peak counts are used.

There are two methods to determine the load pattern in EN 13445. the simplified cycle calculation and the “pool” cycle calculation. Compared to the simplified calculation, the “pool” calculation is more accurate.

The actual vessels cannot bear the absolute static load. In some cases, the cycle number of the vessel is not so big, or the alternative stress amplitude is not so high even with many cycle indexes, so for these vessels, the fatigue analysis can always be ignored. Therefore, in ASME VIII-2, there are three methods introduced to determine whether the fatigue analysis can be ignored, but in EN 13445, there is none. The simplified assessment method in chapter 17 is relatively conservative and can be viewed as a kind of compromise between the complex detailed assessment and the simplified fatigue sieving method. If an accurate result is needed, a detailed assessment method can be adopted.

### 3.6. Linear Fatigue Damage Cumulation

Multi-amplitude stress histories are reduced to a sequence of single-amplitude stress cycles with cycles of equal relevant parameters conveniently together. For each single-amplitude stress cycle, the allowable number of cycles can then be determined. The quotient of the number of cycles of a specific group and the corresponding number of allowable cycles is defined as the fatigue damage index (for this of single-amplitude cycles). For the determination of the cumulative fatigue damage index, the frequently used Palmgren–Miner rule is prescribed.
(3)$$D=\frac{n_{1}}{N_{1}}+\frac{n_{2}}{N_{2}}+\frac{n_{3}}{N_{3}}+\cdots +\frac{n_{k}}{N_{k}}=\sum_{i=1}^{k}\frac{n_{i}}{N_{i}}$$
where *n_i_* is the number of times each stress action is to be subjected, and *N_i_* is the number of permissible load cycles from the fatigue design curve; If D ≤ 1, then this design is acceptable.

### 3.7. Strength Theory

In three methods of ASME VIII-2, the fourth strength theory, namely the Huber–Mises criterion, is used to assess fatigue. However, it is suggested to adopt either the third strength theory, namely the Tresca criterion, or the Huber–Mises criterion to assess the fatigue in EN 13445.

## 4. Finite Element Analysis of Opening Tubing Connection

### 4.1. Structure Dimension and Design Parameter

Models here are from the tests [15]. The structure dimension and the fatigue test are listed orderly in Table 3. For the convenience of explanation, the models will be renumbered.

The model in Table 3 is the abutting nozzle with the full penetration weld. The structure is shown in Figure 3, and the corresponding size is listed in Table 3.

In Table 4, there are two types of abutting nozzle with the full penetration weld. The difference between them is whether the inner wall connecting the nozzle and the cylinder is rounding off or not. The first set (Models 23–29) are right angle structures, and the second sets (Models 30–32) are fillet structures. Their structures are shown in Figure 4.

In Table 5, the models, the embedded structure and the butt welding, away from the discontinuous area of the structure, are applied. The structure is shown in Figure 5. The material properties of all models are listed in Table 6.

### 4.2. Finite Element Model

#### 4.2.1. Models and Grids

Based on the symmetry of the model and load, the 1/4 solid model is established, and the length of the cylinder and the nozzle is much longer than the attenuation length of the edge stress. The SOLID 186 element is adopted to the mesh generation, and the mesh for the weld and the adjacent region is densified. Moreover, in order to use three-point quadratic extrapolation to calculate the hot-spot stress of the weld toe in Method 4, the size of the grid near the expected fatigue hot spot is taken as 0.2 t. The accuracy of the finite element analysis results is not only related to the correctness of load and boundary conditions but also to the number and type of mesh of the finite element model. Therefore, it is necessary to verify the mesh independence and mesh convergence of the finite element model of the tube connections. Three groups of data are selected in this paper. Model 1 is taken as an example. The number of grid nodes and test results are shown in Table 7. Taking Models 1 and Moder 33 as examples, the solid model and grid generation are shown in Figure 6 and Figure 7. Taking Model 1 and Model 33 as an example, the solid model and the mesh generation are shown in Figure 6 and Figure 7.

According to the comparison of the results in the table above, the difference between the mesh test results of different densities is within the allowable error range. Therefore, the grid division of the model meets the requirements of grid independence, and it is considered that the grid has no influence on the calculation results. Considering the accuracy and calculation time, the calculation model selected the mesh with 267,338 nodes and 58,412 elements.

#### 4.2.2. Loads and Boundary Conditions

(1)Apply the symmetry constraint to the symmetry plane.(2)In order to avoid the global displacement of the model, the displacement of the Y-direction at the end face of the nozzle is restricted.(3)Apply the pressure to the internal surface of the model.(4)Apply the equivalent end force to the shell end: The end force is expressed as the uniform pressure acting on the shell end, which is equal to the axial force generated by the internal pressure at the shell end face, divided by the cylinder cross-sectional area.

### 4.3. Finite Element Calculation Results

#### 4.3.1. Elastic Stress Calculation

By taking Model 1 as an example, the calculated elastic stress under *P*_min_ is designated as Load 1 and under *P*_max_ as Load 2. The stress range is designated as Load 3 and expressed as follows:

Load 3 = Load 2 − Load 1

The calculation result of the stress range of Model 1 is shown in Figure 8. It can be seen that the maximum point is at the joint between the cylinder and the internal surface of the nozzle, with a maximum value of 492.1 MPa.

#### 4.3.2. Elastic–Plastic Stress Calculation

Unlike the elastic stress calculation, the material property is set as the multi-linear kinematic hardening in the elastic–plastic stress calculation and adopts the stable stress–strain cyclic curve. According to the finite element analysis, it is determined whether to come to the elastic yield point is through the calculation of Equation (4) based on [20].
(4)$$\sigma_{\mathrm{yield}}=K_{\mathrm{css}}{\left(\varepsilon_{\mathrm{offset}}\right)}^{n_{\mathrm{css}}}$$
where *σ*_yield_ is the yield stress, *K*_css_ is the material parameter in cyclic stress–strain curve mode, *ε*_offset_ is the offset strain and *n*_css_ is the material parameter of cyclic stress–strain curve mode. *ε*_offset_, *K*_css_ and *n*_css_ are obtained from Appendix 3-D of ASME VIII-2; see Table 8 for details.

Then the cyclic stress–strain curve is transformed into the following form:(5)$$\left\{ \begin{array}{l} \varepsilon_{\mathrm{ta}}=\frac{\sigma_{a}}{E_{\mathrm{ya}}}\;\sigma_{a}\leq \sigma_{\mathrm{yield}}\\ \varepsilon_{\mathrm{ta}}=\frac{\sigma_{a}}{E_{\mathrm{ya}}}+{\left[\frac{\sigma_{a}}{K_{\mathrm{css}}}\right]}^{\frac{1}{n_{\mathrm{css}}}}-\varepsilon_{\mathrm{offset}}\;\sigma_{a}>\sigma_{\mathrm{yield}} \end{array}\right.$$
where *ε*_ta_ is the total strain amplitude, *σ*_a_ is the total stress amplitude and *E*_ya_ is the elastic modulus at the corresponding temperature.

For the two-step yield method, the relationship between the plastic strain range and the stress range is expressed as follows:(6)$$\varepsilon_{\mathrm{pr}}=2{\left[\frac{\sigma_{r}}{2K_{\mathrm{css}}}\right]}^{\frac{1}{n_{\mathrm{css}}}}-2\varepsilon_{\mathrm{offset}}$$
where *ε*_pr_ is the plastic strain range, and *σ_r_* is the total stress range.

Equation (6) is fit by MATLAB, and the corresponding point is extracted, as seen in Table 9, which is used to determine the material properties in ANSYS.

The equivalent stress and plastic strain distributions from the finite element modeling of Model 1 are shown in Figure 9 and Figure 10, respectively. It can be seen that the maximum value of the equivalent stress is 448.9 MPa, and the maximum calculated value of the equivalent plastic strain is 0.001678.

#### 4.3.3. Stress Intensity Analysis

In order to evaluate protection against plastic collapse, the results from an elastic stress analysis of the component subject to defined loading conditions are categorized and compared to an associated limiting value. The three basic equivalent stress categories and associated limits that are to be satisfied for plastic collapse are defined in Figure 1. The general primary membrane stress, local primary membrane stress, primary bending stress, secondary stress and peak stress used for elastic analysis are also defined in Figure 1.

The path is shown in Figure 11, and the stress evaluation results are shown in Table 10.

## 5. Fatigue Evaluation Result and Analysis

In order to compare the test life with the allowable cyclic numbers and have a better analysis of the four methods, the coefficient *β* is defined as Equation (7).
(7)$$\beta =\frac{n}{N}$$
where *n* is the cyclic number of the test load, and *N* is the allowable cyclic number. The results are shown in Table 11.

*MAPE* is the mean absolute percentage error, which can represent the deviation between two sets of data.
(8)$$MAPE=\frac{1}{m}\sum_{i=1}^{m}\left|\frac{\beta_{b}-\beta_{o}}{\beta_{b}}\right|$$
where *m* is the sample numbers, *β*_b_ is the dataset of the selected benchmark method, *β*_o_ is the dataset of the other method.

*ω* is the standard deviation, which can represent the degree of dispersion of data. CV is the coefficient of variation, which can represent the fluctuation of data.
(9)$$\left\{ \begin{array}{l} CV=\frac{\omega }{\bar{\beta }}\\ \omega =\sqrt{\frac{1}{m}\sum_{i=1}^{m}{\left(\beta_{i}-\bar{\beta }\right)}^{2}} \end{array}\right.$$
where? $\bar{\beta }$ is the average of *β*.

### 5.1. Evaluation Results of Table 3

Based on the calculation results in Table 3, the values of the coefficient *β* are calculated and shown in Figure 12.

It can be seen that the coefficient *β* of the four methods is larger than 1, which means that the test life is longer than the prospective fatigue life. Therefore, the four methods are all satisfied with the requirement of fatigue strength with a certain safety allowance. It can be seen in Table 12, *ω*_Method 4_ < *ω*_Method 2_ < *ω*_Method 1_ < *ω*_Method 3_, $\bar{\beta }$_Method 4_ < $\bar{\beta }$_Method 2_ < $\bar{\beta }$_Method 1_ < $\bar{\beta }$_Method 3_, and *CV*_Method 3_ < *CV*_Method 4_ < *CV*_Method 2_ < *CV*_Method 1_. From the perspective of discreteness, Method 3 has a greater degree of discreteness than the other three methods, while the other three methods have a similar degree of discreteness, and Method 4 has the smallest discreteness. From the perspective of volatility, Method 3 is more stable than the other three methods, but it is not enough to judge the merits of the method only from the perspective of discreteness and volatility. Compared with other methods, the $\bar{\beta }$ of Method 3 is the largest, that is, the expected fatigue life deviates greatly from the experimental life, so Method 3 is not considered. For the other three methods, the $\bar{\beta }$ of Method 1, Method 2 and Method 4 are close, and $\bar{\beta }$, *ω* and *CV* in Method 4 is smaller than those in Method 1 and Method 2, so Method 4 is the most suitable method in Table 3. The deviation degree between Method 4 and the other three methods is shown in Table 13.

### 5.2. Evaluation Results of Table 4

Based on the calculation results of Table 4, the values of the coefficient *β* are also calculated and shown in Figure 13.

It can be seen that the coefficient *β* of the four methods is larger than 1, which means that the test life is longer than the prospective fatigue life. Therefore, the four methods are all satisfied with the requirement of fatigue strength with a certain safety allowance. The value of β is almost the same in the methods except Method 3. It also shows that whether the inner angle of the connection between the cylinder and the nozzle is chamfer or not has no significant influence on the coefficient. It can be seen from Table 14 that *ω*_Method 2_ < *ω*_Method 4_ < *ω*_Method 1_ < *ω*_Method 3_, $\bar{\beta }$_Method 4_ < $\bar{\beta }$_Method 2_ < $\bar{\beta }$_Method 1_ < $\bar{\beta }$_Method 3_, and *CV*_Method 2_ < *CV*_Method4_ < *CV*_Method 1_ < *CV*_Method 3_. $\bar{\beta }$, *ω* and *CV* in Method 3 are large in value, which makes it not considered. The values of $\bar{\beta }$ in Method 1, Method 2 and Method 4 are close, but there is a big difference in the values of *CV* and *ω* among the three, where *ω* and *CV* in Method 2 are the smallest. Consequently, Method 2 is the most suitable method in Table 4. The deviation degree between Method 2 and the other three methods is shown in Table 15.

### 5.3. Evaluation Results of Table 5

Similarly, based on the calculation results of Table 5, the values of the coefficient *β* are calculated and shown in Figure 14.

The models in Table 5 are all the embedded nozzle structures with the butt weld. It can be seen from Figure 14 that the coefficient *β* of the four methods is larger than 1, which means that the test life is longer than the prospective fatigue life. In this way, the four methods are all satisfied with the requirement of fatigue strength with a certain safety allowance. It can be seen from Table 16 that *ω*_Method 4_ < *ω*_Method 2_ < *ω*_Method 3_ < *ω*_Method 1_, $\bar{\beta }$_Method 4_ < $\bar{\beta }$_Method 2_ < $\bar{\beta }$_Method 3_ < $\bar{\beta }$_Method 1_, and *CV*_Method 4_ < *CV*_Method 3_ < *CV*_Method 2_ < *CV*_Method 1_. The $\bar{\beta }$, *ω* and *CV* in Method 4 are all smaller than the other three methods, so Method 4 is the best one in Table 5. The deviation degree between Method 4 and the other three methods is shown in the table below. Both Method 1 and Method 3 have large deviations. Compared to the above models, the embedded nozzle structure is the butt weld, and the weld position is away from the structural discontinuity regions. For this kind of structure, the welding position is not evaluated in Method 1 (The most dangerous point is located at the inner corner of the discontinuous region of the nozzle). In Method 3, the welding position is calculated, but it is suggested to use the welded joint of smooth contour without mechanical processing. This may be the reason why the results between Method 1 and Method 3 are so different. The deviation degree between Method 4 and the other three methods is shown in Table 17.

### 5.4. Result Analysis

Based on the analysis, Method 4 is the most accurate and reliable approach by acquiring the hot-spot stress of the potential fatigue location through extrapolation. Method 2 is for the elastic–plastic analysis, which concerns the strain strengthening when the stress exceeds the yield strength during the transformation of the materials under the load. Therefore, the result of Method 2 is more accurate than that of Method 1 by the elastic analysis. Compared to the other methods, Method 3 is more conservative. That is maybe because the welded joint in the test structure is smoothed, and this structure is not applied to Method 3.

The elastic–plastic analysis is very complicated; it requires the stable stress–strain cyclic curve of the material, which is presented in the form of the fitting formula. The curve can be obtained from the material test or even more conservative than the specified cycle of the materials. It is suggested that Method 1 is the preference for most components, while Method 2 is adopted for some key components or some severe conditions.

The equivalent structure stress range is used in the welded material. This stress is the function of the membrane stress and the bending stress perpendicular to the hypothetical potential crack. This method is only applied to the assessment of the welded joint. The detailed fatigue assessment separately adopts the notch stress and the hot-spot stress for the unwelded and the welded structure, which can be regarded as the combination of Method 3 and Method 1, and the structural stress of the hot-spot obtained by the extrapolation is used as the assessment parameter.

## 6. Conclusions

The fatigue assessment of tube connections under pressure is discussed using the four methods from ASME VIII-2 and EN 13445-3. Based on the comparison to the data of the fatigue test, the following conclusions are obtained:(1)For the calculation of the elastic stress, Method 1 adopts the effective total equivalent stress amplitude for assessing the fatigue damage. Method 1 is the most widely used traditional method and can be used in both welded structure and un-welded structures. This method has simple operation, safety and reliability.(2)For the elastic–plastic calculation, Method 2 adopts the effective strain range for assessing the fatigue damage and can be used in both the welded structure and the unwelded structure. This method is with high accuracy, good stability, safety and reliability, but it is difficult to obtain the stable stress–strain cyclic curves of the corresponding materials. Furthermore, the elastic–plastic analysis is very complicated.(3)For the calculation of the elastic stress, Method 3 adopts the equivalent structure stress for assessing the fatigue damage. This method is applied for the fatigue assessment of the welded. It is suggested to be used for the welded joint without mechanical processing. This method is developed under fracture mechanics, but it is still conservative and unstable, and the procedure is very complicated.(4)In Method 4, the detailed assessing procedure is performed separately for the welded and unwelded. For the welded, Method 4 applies the hot-spot stress obtained from the principal stress by the extrapolation for the assessment. For the unweldment, it applies the notch stress as the assessment parameter. The iterative calculations are required. This method is the most accurate, stable and reliable.

## Figures and Tables

**Figure 1 materials-16-00231-f001:**
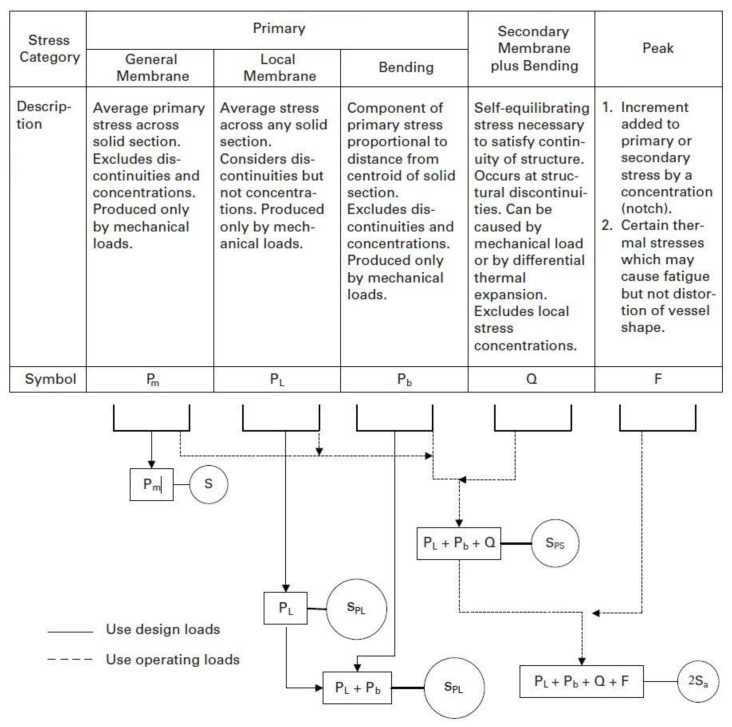
Stress categories and limits of equivalent stress.

**Figure 2 materials-16-00231-f002:**
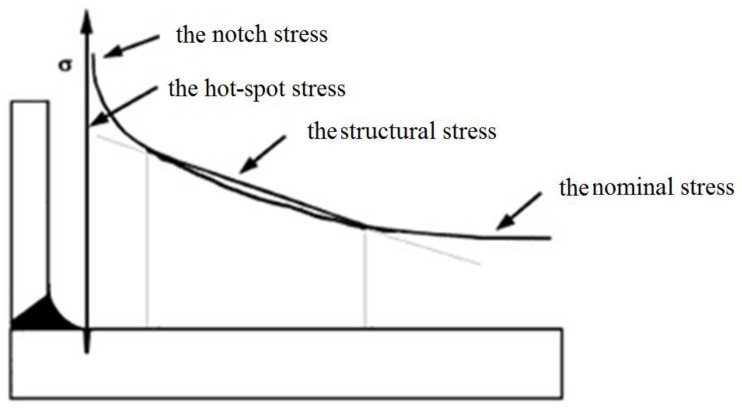
Stress type and location schematic.

**Figure 3 materials-16-00231-f003:**
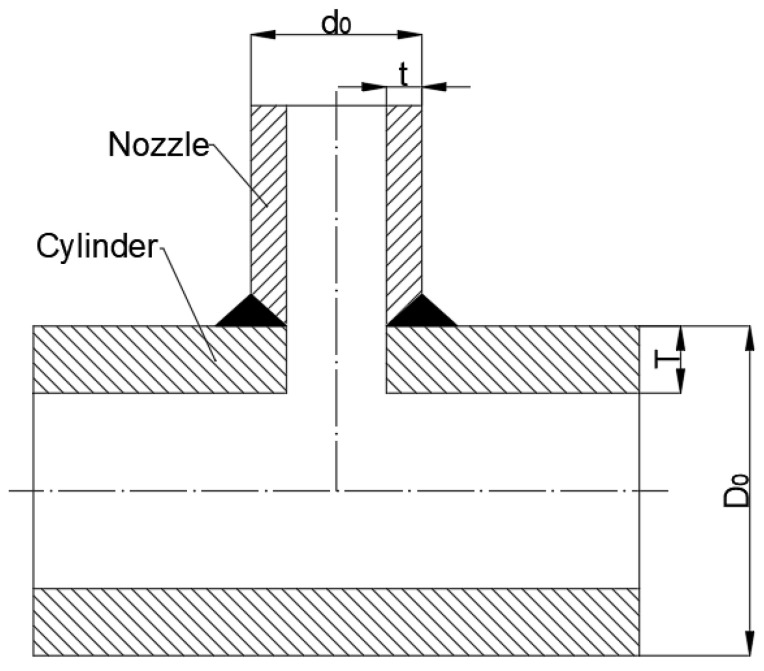
Structure of specimen (Table 3).

**Figure 4 materials-16-00231-f004:**
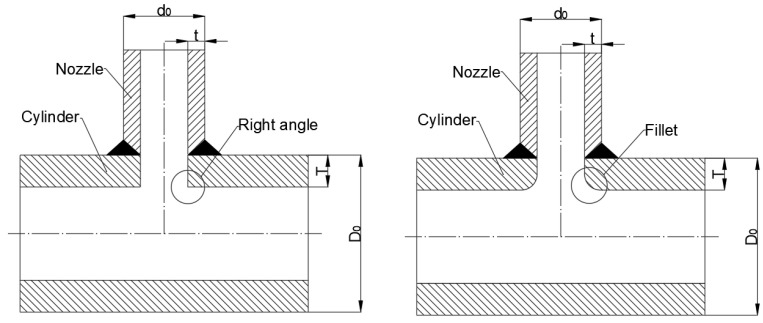
Structure of specimen (Table 4).

**Figure 5 materials-16-00231-f005:**
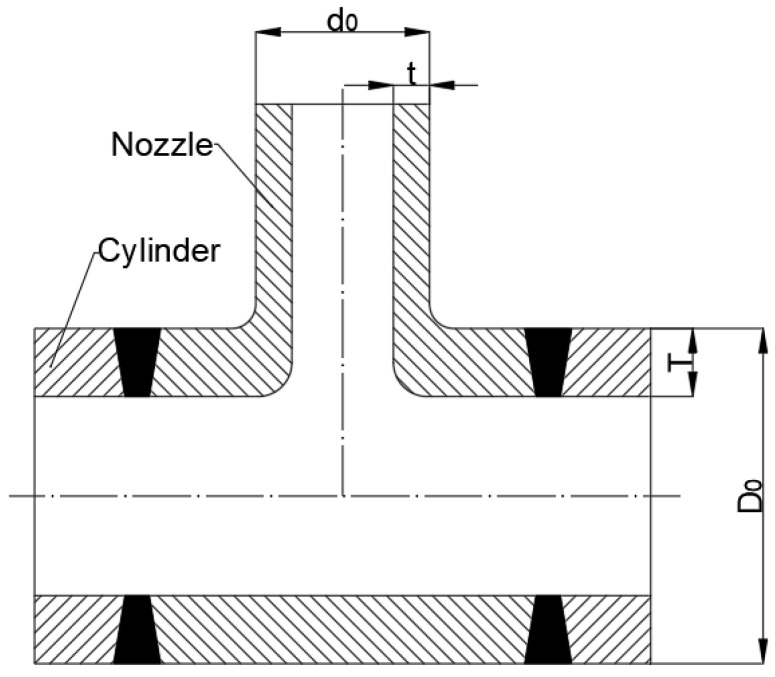
Structure of specimen (Table 5).

**Figure 6 materials-16-00231-f006:**
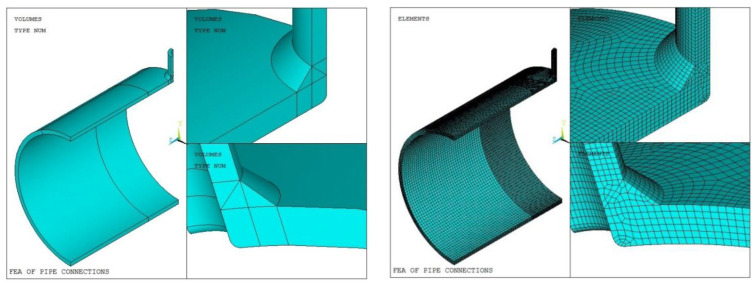
Solid model and mesh generation of Model 1.

**Figure 7 materials-16-00231-f007:**
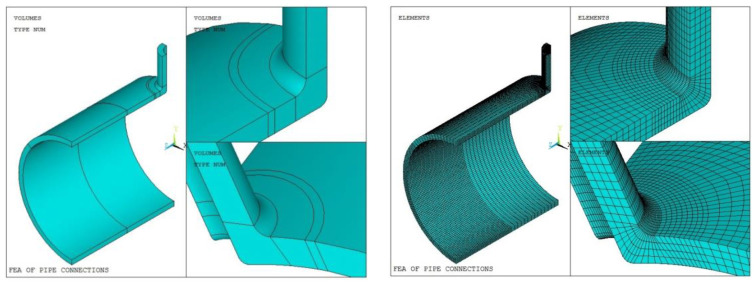
Solid model and mesh generation of Model 33.

**Figure 8 materials-16-00231-f008:**
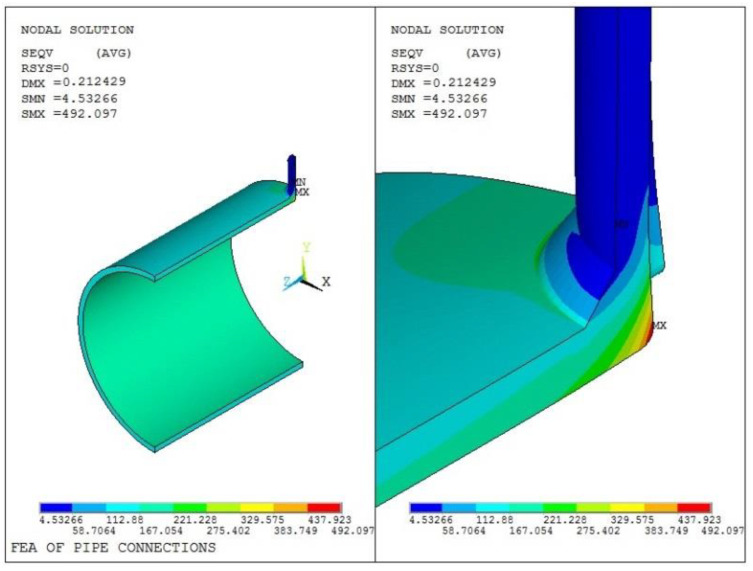
Huber–Mises stress analysis result of Model 1.

**Figure 9 materials-16-00231-f009:**
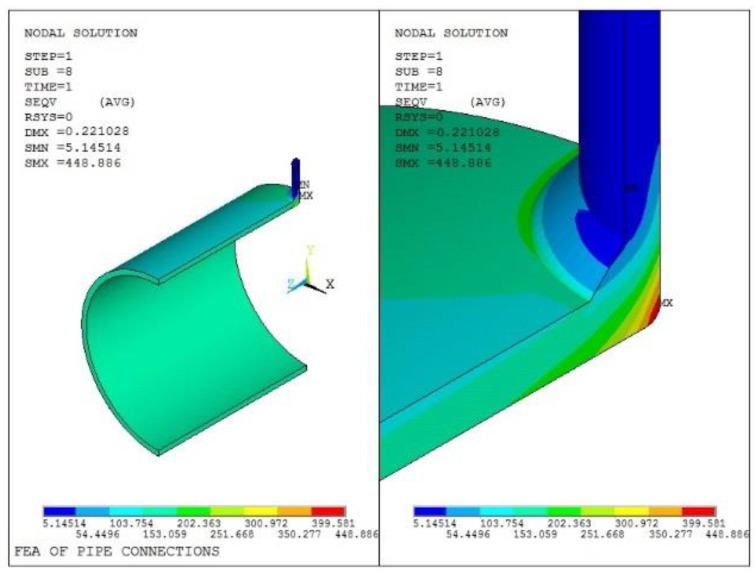
Equivalent stress analysis result of Model 1.

**Figure 10 materials-16-00231-f010:**
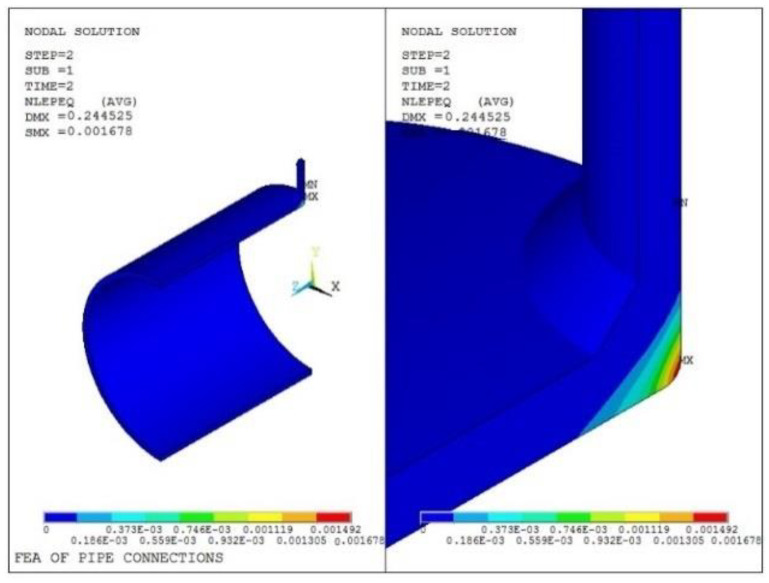
Equivalent plastic strain stress result of Model 1.

**Figure 11 materials-16-00231-f011:**
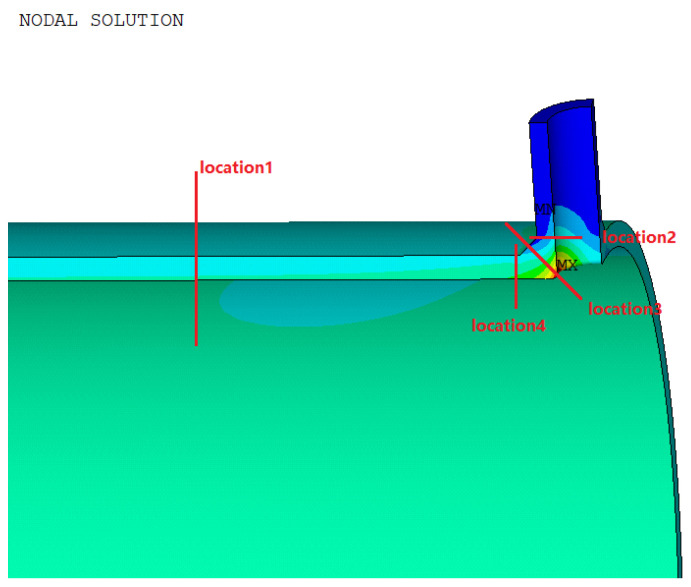
Path assessment chart of Model 1.

**Figure 12 materials-16-00231-f012:**
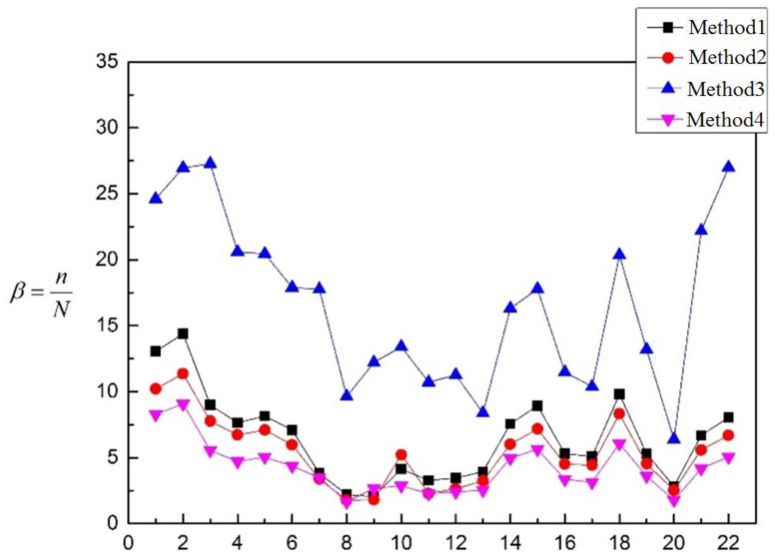
Fatigue evaluation results of models in Table 3.

**Figure 13 materials-16-00231-f013:**
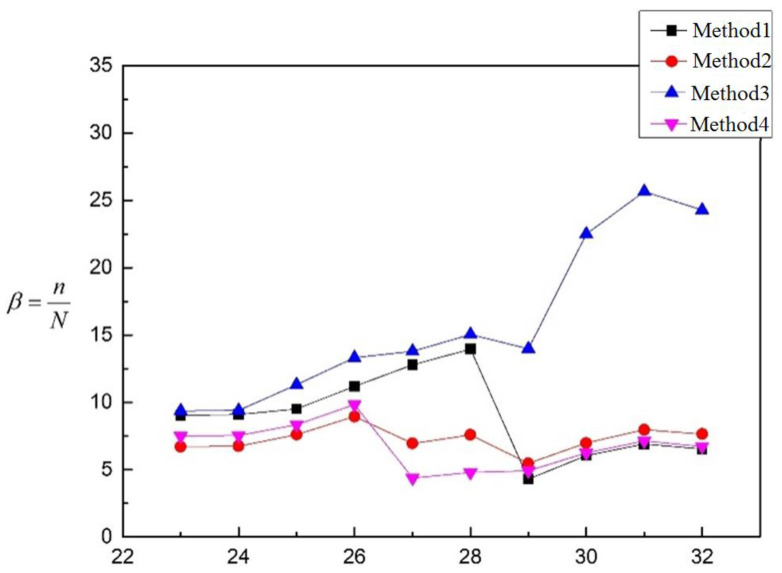
Fatigue evaluation results of models in Table 4.

**Figure 14 materials-16-00231-f014:**
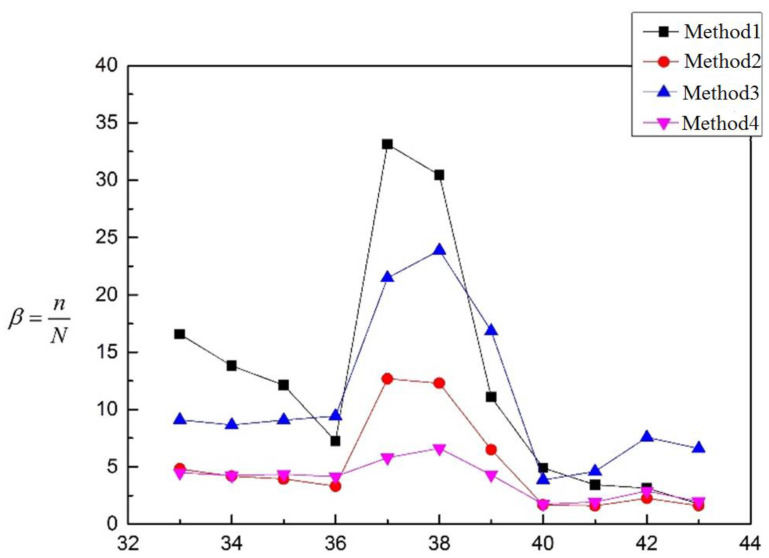
Fatigue evaluation results of models in Table 5.

**Table 1 materials-16-00231-t001:** Load case combinations and allowable stresses for an elastic analysis.

Design Load Combination	Allowable Stress
(1) P + Ps + D (2) P + Ps + D + L (3) P + Ps + D + L + T (4) P + Ps + D + Ss (5) 0.6 D + (0.6 W or 0.7 E) (6) 0.9 P + Ps + D + (0.6 W or 0.7 E) (7) 0.9 P + Ps + D + 0.75 (L + T) + 0.75 Ss (8) 0.9 P + Ps + D + 0.75 (0.6 W or 0.7 E) + 0.75 L + 0.75 Ss	Determined based on the stress category shown in Figure 1

NOTES: Loads listed herein shall be considered to act in the combinations described above, whichever produces the most unfavorable effect in the component being considered. Effects of one or more loads not acting shall be considered.

**Table 2 materials-16-00231-t002:** Load descriptions.

Design Load Parameter	Description
L	Appurtenance live loading Effects of fluid momentum, steady state and transient Loads resulting from wave action
E	Earthquake loads
W	Wind loads
W_pt_	The pressure test wind load case. The design wind speed for this case shall be specified by the owner user.
S_s_	Snow loads
T	The self-restraining load case (i.e., thermal loads, applied displacements). This load case does not usually affect the collapse load but should be considered in cases where the elastic follow-up causes stress to be insufficient to redistribute the load without excessive deformation.

**Table 3 materials-16-00231-t003:** Structure dimensions and test data.

NO.	Cylinder Diameter	Cylinder Thickness	Nozzle Diameter	Nozzle Thickness	Testing Pressure		Testing Cycle Life
	*Do*/mm	*T*/mm	*do*/mm	*t*/mm	*P*_max_/MPa	*P*_min_/MPa	n
1	325	10	44.2	7	1.2931	*P*_min_ *=* 0.1 *P*_max_	180,200
2	325	10	44.2	7	1.3272		182,100
3	325	10	140.2	11	1.1223		150,900
4	325	10	159.5	15.9	1.3272		104,800
5	325	10	159.3	22	1.4199		155,000
6	325	10	159.3	22	1.376		149,700
7	325	10	252	22	1.1174		90,100
8	325	10	252	22	1.1711		42,100
9	325	10	268.3	32	1.1711		79,600
10	600	20	303.7	34	1.3367		79,400
11	600	20	303.7	34	1.2839		72,000
12	600	20	303.7	34	1.2997		72,800
13	325	20	44.2	7	2.7821		41,200
14	325	20	44.2	7	2.5503		105,400
15	325	20	159.3	22	2.2075		171,200
16	325	20	159.3	22	2.3184		86,400
17	325	20	244.6	22	2.3083		43,300
18	325	20	244.6	22	2.0664		119,900
19	325	20	267.5	32	2.1067		97,700
20	325	20	267.5	32	1.7338		95,400
21	325	20	139.1	11	1.9959		91,000
22	325	20	139.1	11	1.9051		128,000

**Table 4 materials-16-00231-t004:** Structure dimensions and test data.

NO.	Cylinder Diameter	Cylinder Thickness	Nozzle Diameter	Nozzle Thickness	Testing Pressure		Testing Cycle Life
	*D*o/mm	*T/*mm	*do/*mm	*t/*mm	*P*_max_/MPa	*P*_min_/MPa	n
23	342.90	19.050	50.700	9.525	25.389	0	51,000
24					25.389		51,300
25					22.712		87,500
26					22.712		103,000
27					46.884		9990
28					46.884		10,900
29					39.341		18,800
30					34.149		46,900
31					34.149		53,500
32					34.474		49,100

Note: the fillet diameter of the second set is 3.175 mm.

**Table 5 materials-16-00231-t005:** Structure dimensions and test data.

NO.	Cylinder Diameter	Cylinder Thickness	Nozzle Diameter	Nozzle Thickness	Testing Pressure		Testing Cycle Life
	*D*o/mm	*T/*mm	*do/*mm	*t/*mm	*P*_max_/MPa	*P*_min_/MPa	n
33	321.4	13	70.9	13	37.296	0	6500
34					35.317		7600
35					33.339		9800
36					29.441		15,600
37	321.4	13	94.8	13	39.216		8600
38					35.317		14,800
39					29.441		20,500
40	334	20	110.1	20.8	49.010		3460
41					43.125		6400
42					39.261		14,300
43					33.376		20,900

**Table 6 materials-16-00231-t006:** Material properties.

Model		Structure	Material	Allowable Stress S/MPa	Yield Strength Sy/MPa	Tensile Strength Su/MPa	Elastic Modulus E/×10^3^ MPa	Poisson’s Ratio
Table 3	1–22	Cylinder	Carbon Steel	180	283	434	201	0.3
		Nozzle		174	262	446		
Table 4	23–26	Cylinder Nozzle	A201A	162	244	388		
	27–32		A302B	275	533	660		
Table 5	33–36 40–43	Cylinder	FTW60	285	617	686		
		Nozzle	JISSF60	235	353	597		
	37–39	Cylinder	FTW60	285	617	686		
		Nozzle	302B	269	496	647		

**Table 7 materials-16-00231-t007:** Mesh convergence results.

Number of Grid Nodes	Number of Elements	Maximum Stress/MPa	Deviation
135,117	27,856	518.724	5.41%
267,338	58,412	492.097	datum
398,546	88,660	499.350	1.47%

**Table 8 materials-16-00231-t008:** Parameters and the yield point of the cyclic stress–strain curve of Model 1.

Material	*ε* _offset_	*K*_css_/MPa	*n* _css_	*σ*_yield_/MPa
Carbon Steel	2.0 × 10^−5^	757	0.128	189.508
Carbon Steel-Welded	2.0 × 10^−5^	695	0.110	211.397

**Table 9 materials-16-00231-t009:** Plastic material properties of Model 1.

NO.	Carbon Steel		Carbon Steel-Welded	
	Stress Range/MPa	Plastic Strain Range	Stress Range/MPa	Plastic Strain Range
1	379.0161	0	422.7933	0
2	400	0.000020938	450	0.000030515
3	450	0.00011294	500	0.00014376
4	500	0.00030834	550	0.00039708
5	550	0.00069347	600	0.00092403
6	600	0.0014	650	0.0020
7	650	0.0027	700	0.0039
8	700	0.0048	750	0.0073
9	750	0.0082	800	0.0131
10	800	0.0137	850	0.0228

**Table 10 materials-16-00231-t010:** Stress evaluation results of Model 1.

**Stress Evaluation Area**	**P_m_/MPa**	**(P_m_ + P_b_)** **/MPa**	**S_m_/MPa**	**Stress Evaluation**	**Results**
1	160.64	193.99	180	$P_{L}\;<\;1.0\;S_{m}$ $P_{L}{\;+\;P}_{b}{\;<\;3\;S}_{m}$	qualified
**Stress Evaluation Area**	**P_L_/MPa**	**(P_L_ + P_b_)** **/MPa**	**S_m_/MPa**	**Stress Evaluation**	**Results**
2	84.719	144.38	174	$P_{L}\;<\;1.5\;S_{m}$ $P_{L}{\;+\;P}_{b}{\;<\;3\;S}_{m}$	qualified
3	249.63	464.58	174	$P_{L}\;<\;1.5\;S_{m}$ $P_{L}{\;+\;P}_{b}{\;<\;3\;S}_{m}$	qualified
4	206.19	281.57	180	$P_{L}\;<\;1.5\;S_{m}$ $P_{L}{\;+\;P}_{b\;}{<\;3\;S}_{m}$	qualified

**Table 11 materials-16-00231-t011:** Summary of fatigue evaluation results.

Model NO.	Fatigue Evaluation Result				Model NO.	Fatigue Evaluation Result			
	Method 1	Method 2	Method 3	Method 4		Method 1	Method 2	Method 3	Method 4
1	13.047	10.210	24.600	8.270	23	9.032	6.705	9.351	7.492
2	14.367	11.354	26.973	9.072	24	9.085	6.744	9.406	7.536
3	8.979	7.764	27.279	5.550	25	9.499	7.600	11.312	8.342
4	7.642	6.718	20.588	4.699	26	11.182	8.946	13.316	9.820
5	8.147	7.085	20.444	5.0374	27	12.784	6.945	13.790	4.380
6	7.081	5.969	17.887	4.367	28	13.948	7.578	15.046	4.779
7	3.805	3.384	17.775	3.483	29	4.290	5.455	13.972	4.910
8	2.188	1.776	9.625	1.658	30	6.041	6.974	22.492	6.241
9	2.038	1.799	12.213	2.662	31	6.891	7.956	25.657	7.119
10	4.132	5.212	13.391	2.856	32	6.533	7.639	24.284	6.705
11	3.272	2.281	10.716	2.268	33	16.575	4.820	9.1058	4.499
12	3.448	2.635	11.255	2.388	34	13.805	4.195	8.662	4.251
13	3.924	3.237	8.369	2.534	35	12.129	3.940	9.070	4.358
14	7.544	6.015	16.308	4.957	36	7.257	3.309	9.441	4.157
15	8.923	7.174	17.788	5.631	37	33.133	12.675	21.466	5.796
16	5.305	4.504	11.461	3.347	38	30.434	12.295	23.892	6.613
17	5.075	4.421	10.389	3.122	39	11.086	6.497	16.861	4.306
18	9.803	8.320	20.345	6.049	40	4.921	1.680	3.867	1.758
19	5.287	4.523	13.189	3.606	41	3.446	1.602	4.609	1.933
20	2.817	2.517	6.369	1.785	42	3.150	2.2617	7.578	2.891
21	6.678	5.581	22.197	4.176	43	1.785	1.611	6.600	1.994
22	8.050	6.676	26.985	5.055					

**Table 12 materials-16-00231-t012:** *ω*, $\bar{\beta }$ and *CV* of the four methods.

	Method 1	Method 2	Method 3	Method 4
*ω*	3.25	2.57	6.27	1.89
$\bar{\beta }$	6.43	5.42	16.64	4.21
*CV*/%	51.63	48.66	38.55	46.06

**Table 13 materials-16-00231-t013:** Method 4 deviation from other methods.

	Method 4 and Method 1	Method 4 and Method 2	Method 4 and Method 3
*MAPE*/%	51.57	30.32	312.58

**Table 14 materials-16-00231-t014:** *ω*, $\bar{\beta }$ and *CV* of the four methods.

	Method 1	Method 2	Method 3	Method 4
*ω*	2.92	0.88	5.75	1.63
$\bar{\beta }$	8.93	7.25	15.86	6.73
*CV*/%	32.70	12.10	36.25	24.14

**Table 15 materials-16-00231-t015:** Method 2 deviation from other methods.

	Method 2 and Method 1	Method 2 and Method 3	Method 2 and Method 4
*MAPE*/%	35.01	119.28	16.01

**Table 16 materials-16-00231-t016:** *ω*, $\bar{\beta }$ and *CV* of the four methods.

	Method 1	Method 2	Method 3	Method 4
*ω*	10.17	3.82	6.38	1.51
$\bar{\beta }$	12.52	4.99	11.01	3.87
*CV*/%	81.20	76.57	57.95	38.95

**Table 17 materials-16-00231-t017:** Method 4 deviation from other methods.

	Method 4 and Method 1	Method 4 and Method 2	Method 4 and Method 3
*MAPE*/%	183.00	32.41	174.19

## Data Availability

Not applicable.

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
