# Peer review of "Investigations on Fatigue Life of Tube Connections Based on International Codes of Pressure Vessel"

_materials, 2022, doi:10.3390/ma16010231_

Round 1

Reviewer 1 Report

The presented work is variable for engineers. Unfortunately, there are many shortcomings, which must be corrected. The following comments are below:

1.     Chapter 3.4 paragraph e) What surface correction is for methods other than method 4?

2.     Chapter 3.6 More details must be given. What method for damage cumulation is adopted?

3.     Figures 3, 4 and 5 A dimensions on the drawing must be added.

4.     Chapter 4.2.1 How many elements and nodes were used?

5.     What value has N?

6.     How do changes in the geometry of the specimen influence the fatigue life? Figure 4 presents the specimen with fillet and right angle. How does it change fatigue life/strength? Figure 5 presents different specimens. It wasn’t presented in Figure 6.

7.     It was the statement that Method 4 is “complicated in the assessment process, it is the most accurate, stable and reliable method”. How does this complication influence the time of the calculation?

Particular comments are the following:

Line 23 Missing reference. Reference should be added after the name of the author, not at the end of the sentence.

Line 26 It should be swapped to a comma.

Lines 216-219 The sentence is hard to understand. Please, rewrite.

Line 238 Is it method 3? In my opinion, it should be method 4.

Line 330 It should be Huber-von Mises stress. The defining equation for the von Mises stress was first proposed by Huber in 1904. Von Mises describes this equation in 1913. Please correct.

Line 385 The stress should be rounded up to a decimal fraction.

Table 7 In my opinion εoffset has a value of 2.0 x 10-3.

Reviewer 2 Report

Title of the Manuscript: Investigations on Fatigue life of Tube Connections Based on International Codes of Pressure Vessel

Reviewer Comments:

The subject of the paper is interesting and paper could be accepted for the possible publication in Materials Journal, MDPI publication. Nevertheless, the paper requires Minor Revision, prior to the publication. The things that need revision (in order of appearances):

1. In Introduction part- Failure modes of the pressure vessel (due to buckling, overload, and fast fracture) need to discuss with relevant literature.

2. In introduction part- At what pressure ASME is required? Need to be discussed.

3. A number of conditions affect fatigue life of a pressure vessel. (For eg. Cyclic stress state, Surface quality, Material type, Residual stresses, Geometry etc.) It would be beneficial if the authors could add a brief description of the same thing.

4. Provide the proper dimensions for the diagrams.- Fig.3, 4 and 5

5. In Table 3, two different cylinder diameter and thickness has been chosen by the author. Reader is curious about on what basis above parameters are selected and what is the impact of above parameters w.r.t authors present research work? Justify.

6. How do you calculate cumulative damage? Need to be incorporate in Chapter 3.6

Reviewer 3 Report

The paper presents results of simulations and fatigue performances of the tube connections by FEA technique. However, there are several critical aspects of the work to be pointed out:

1. The abstract is very general, and there is a lot of irrelevant information. The abstract should be explained and showed the important aspects of paper. So, this abstract in the present form is unacceptable. Standards and software systems must be avoided, it is necessary to highlight the results of the study much more concretely and deeply analyzed.

2. The previous work is medium level, and isn't sufficient. It is recommended updated this section with new references, and compared them with your work.  There are many experimental and simulated works that present the results of tube connection tests.

3. What mean stress theory did you use for fatigue analysis (S-N, Goodman, Gerber…)?

4. FEA: What is the type of mesh? Mesh convergences should be involved, the optimal element and size element should be specified.

5. A thorough analysis of the equivalent stress must be performed. The stresses that appear in the FEA model must be compared with the strength of the materials from which the tubes are made.

6. “The comparison of fatigue life between numerical simulation and experimental results is discussed”. I did not find any experimental data in the paper. Fatigue testing of the tubes is recommended to validate the FEA simulations.

7. The novelty of the paper should be emphasized compared to other contributions in this field in each section appropriately. So, when the numerical simulation is presented, it should be written in a detailed way that what is new compared to other authors’ example. How this example can be implemented in practice etc. Overall, the analysis and discussion part of this paper should be deeper.

8. As the data analysis of this article, more statistical analysis (CV, standard deviation) should been added, for example, the coefficient of variation of the simulation data should been added and discussed. This lack of statistics is not acceptable in professional publications and needs to be completed.

Round 2

Reviewer 1 Report

Accept, after minor correction. Fig 8-9 should have in legend Huber-Mises as it is in the text.

Reviewer 3 Report

I recommend publishing the paper.

Author Response

Thank you!